# A New Strategy for Nucleic Acid Delivery and Protein Expression Using Biocompatible Nanohydrogels of Predefined Sizes

**DOI:** 10.3390/pharmaceutics15030961

**Published:** 2023-03-16

**Authors:** Lakshmanan Eswaran, Gila Kazimirsky, Ronen Yehuda, Gerardo Byk

**Affiliations:** Laboratory of Nanobiotechnology, Department of Chemistry, Bar-Ilan University, Ramat Gan 52900, Israel

**Keywords:** self-assembly, polymerization, cationic nanohydrogels, non-viral gene delivery

## Abstract

We have developed new formulations of nanohydrogels (NHGs) complexed with DNA devoid of cell toxicity, which, together with their tuned sizes, makes them of great interest for delivering DNA/RNA for foreign protein expression. Transfection results demonstrate that, unlike classical lipo/polyplexes, the new NHGs can be incubated indefinitely with cells without apparent cellular toxicity, resulting in the high expression of foreign proteins for long periods of time. Although protein expression starts with a delay as compared to classical systems, it is sustained for a long period of time, even after passing cells without observation of toxicity. A fluorescently labelled NHG used for gene delivery was detected inside cells very early after incubation, but the protein expression was delayed by many days, demonstrating that there is a time-dependent release of genes from the NHGs. We suggest that this delay is due to the slow but continuous release of DNA from the particles concomitantly with slow but continuous protein expression. Additionally, results obtained after the in vivo administration of m-Cherry/NHG complexes indicated a delayed but prolonged expression of the marker gene in the tissue of administration. Overall, we have demonstrated gene delivery and foreign protein expression using GFP and m-Cherry marker genes complexed with biocompatible nanohydrogels.

## 1. Introduction

Gene therapy has emerged as an essential technique to treat genetically caused diseases, such as cancer, Alzheimer’s disease (AD), Parkinson’s disease (PD), and spinal muscular atrophy (SMA) [1,2,3,4]. Nevertheless, the transfer of therapeutic nucleic acids in the form of plasmids for gene expression, siRNA, and antisense oligonucleotides for gene silencing is still a difficult challenge [5,6,7]. Nowadays, the most efficient systems for gene delivery are still viral carriers, which have been successfully applied and used in clinical trials. However, they have some essential drawbacks, such as immunogenicity, the limited size of the therapeutic gene that can be packed in the virus, and the risk of recombination with latent inactive virus homing in the host having already been demonstrated [8,9]. In contrast, non-viral carriers have many benefits, such as lack of immunogenicity, unlimited DNA loading capacity, higher chemical versatility, and high biocompatibility, if properly designed [10]. Many kinds of non-viral systems have been employed for gene delivery, such as polyethyleneimine (PEI) [11], poly(*L*-lysine) (PLL) [12], (PAMAM) and its dendrimer [13,14,15], poly(*L*-glutamic acid) [16], polyphosphoester [17] chitosan, poly(2-dimethylamino) ethyl methacrylate (PDMAEMA) [18], lipids, and inorganic nanoparticles [19,20,21]. PEI and other cationic polymers have a higher transfection efficiency when complexed with nucleic acids. However, the drawbacks of using PEI polymers include high toxicity and poor stability in gene delivery to the immune system [22]. Among all the non-viral systems, cationic polymeric nanoparticle systems have gained much interest due to safety concerns and the relatively high gene transfection efficiencies [23]. In general, cationic nanoparticles can form polyplexes with negatively charged nucleic acids, including plasmid DNA, siRNA, and mRNA, through electrostatic interactions. Polymeric nanoparticles with subcellular sizes can be loaded with “cargo” molecules and effectively endocytosed by cells, resulting in high cellular uptake of the cargos. In the last few decades, there has been considerable work on the development and application of biocompatible polymeric nanoparticles as nucleic acid delivery systems [24], especially in the field of delivery of biomolecules, such as DNA, RNA, and proteins [25,26,27]. Polymeric nanoparticles for effective bio-application need to possess several properties, such as biocompatibility, high stability in non-delivery areas, adjustable particle size, high cargo loading, and short-term cargo release. Many non-viral systems still result in low transient gene expression in target cells due to uncontrollable DNA degradation profiles, resulting in the rapid loss of unintegrated genes. Thus, limited transfection efficiency is a challenge that needs repeated administrations of therapeutic genes to maintain considerable therapeutic levels, which can cause unnecessary pain and place economic burdens on patients [28]. In this study, we have developed new DNA formulations of a novel biocompatible NHG system recently demonstrated to be able to complex DNA [29] with improved biological properties for gene delivery.

## 2. Materials and Methods

### 2.1. Materials

An XTT Assay Kit was purchased from Biological Industries, Beit Haemeq, Israel. A humidified incubator (Water-Jacketed, US Autoflow Automatic CO_2_ Incubator manufactured by NuAire, Inc., Plymouth, MN, USA) was used for cell culture. In vitro studies were carried out using fluorescence microscopy (Leica, TIRF) and an Incucyte^®^ SX5 controller (Essen BioScience, A Sartorius company, Ann Arbor, MI, USA). In vivo studies were analyzed by the Maestro II in vivo imaging system with 2D planar fluorescence imaging of small animals (Cambridge Research and Instrumentation (CRi), Inc., Woburn, MA, USA). A yellow excitation/emission filter set was used for our experiments (λex, 575.5–620.5 nm; λem > 630 nm). The Liquid Crystal Tunable Filter (LCTF) was programmed to acquire image cubes from λ = 630 to 700 nm, with an increment of 10 nm per image. Fluorescence intensity measurements were calculated as average intensity over the surface area using Cri Maestro software and ImageJ software. 

Nanohydrogels of 50, 200 and 400 nm were prepared and characterized as described in a separate publication [29]. Cationic lipid RPR120535 was synthesized as previously described [30].

### 2.2. Synthesis of NHGs

Briefly, the new biocompatible NHGs were obtained by reacting *N*-isopropyl acrylamide (NIPAM), acrylonitrile (ACN), surfactant (PVP), a cross-linker (BIS), and acrylated jeffamine macromonomers mixed at different ratios in the presence of initiator KPS, as reported in [29]. Sizes were well monodispersed, with polydispersity indexes between 0.13 and 0.33, as measured by DLS. Atomic force microscopy confirmed the sizes of the NHGs. The presence of ACN groups in the NHGs allows NHGs with high numbers of free amino groups upon reduction to be obtained. After the obtention of the nitrile-containing NHGs, they were submitted to chemical reduction. The reduction reactions were performed using borane:THF. Reduced NHGs were identified by DLS and atomic force microscopy.

### 2.3. Labelling of NHGs 

A quantity of 1 mg of Cy5-NHS ester was dissolved in 50 µL DMSO and was added to a 4 mL vial containing 1 mL water, 3 mg of 200 nm NHGs, and 10 mg of solid NaHCO_3_. The mixture was stirred overnight in the dark to obtain Cy5@NHGs. The Cy5@NHGs were dialyzed in water for 3 days. The water was changed twice per day. The fluorescence of NHGs was confirmed by a fluorescence microscope (Leica, Wetzlar, Germany, TIRF).

### 2.4. Cell Culture

HeLa, NIH-3T3, and HEK-293T cells were cultured in DMEM medium and supplemented with 10% fetal bovine serum (FBS, Sigma-Aldrich, St. Louis, MO, USA), 100 μg/mL streptomycin, and 100 U/mL penicillin at 37 °C in 5% CO_2_.

### 2.5. Plasmid Preparation 

The plasmids used in this work were GFP and m-Cherry: 1-Monster Green^®^ Fluorescent Protein GFP phMGFP Vector. The latter contains the open reading frame for the Monster Green^®^ Fluorescent Protein cloned into a mammalian expression vector. The Monster Green^®^ Fluorescent Protein is encoded by an improved synthetic version of the green fluorescent protein gene originally cloned from Montastrea cavernosa (Great Star Coral), and the synthetic gene (hMGFP) expresses a 26 kDa protein (source: www.promega.com/tbs; catalog number: E6421). 2-mCherry-N1 is a mammalian expression vector designed to express a protein of interest fused to the *N*-terminus of mCherry, a mutant fluorescent protein derived from the tetrameric *Discosoma* sp. red fluorescent protein, DsRed. The unmodified vector can be used to express mCherry in mammalian cells (source: www.takarabio.com; catalog number: 632523). The plasmid DNA was amplified in E. coli and purified according to the supplier’s protocol (Promega, Madison, WI, USA). The quantity and quality of the purified plasmid DNA were assessed by optical density at 260/280 nm and by electrophoresis in 0.8% agarose gel.

### 2.6. Cell Proliferation Assays 

The proliferation of the cells was analyzed by XTT assay. The cells with their media were seeded in 96-well flat-bottom plates at a density of 8 × 10^3^ cells per well and cultured in a humidified incubator (at 37 °C for 24 h). Then, the media were removed by aspiration, and 90 μL quantities of fresh media were added. The cells were treated with 10 µg of naked NHGs, polyplexes (10 µg NHGs/0.5 µg DNA), lipoplexes (6 nmol/0.5 µg), 0.5 µg DNA, and 2% triton. Briefly, a dose of 10 μL of polyplexes/lipoplexes in medium was added to each well (in quadruplicate) and incubated for 6 days. One row was used as control cells without the addition of polyplexes/lipoplexes, and one row containing only the appropriate medium was used for blank absorbance readings. Subsequently, 50 μL of XTT reagent (with initiator) was added to the cells and incubated for 4 h. The cell absorbance was read at 570 nm in a TECAN microplate reader. Cell proliferation % = (Isample/Icontrol) × 100, where Isample is the absorbance of NHG-treated wells and Icontrol is the absorbance of control wells without NHG treatment. These experiments were repeated twice.

### 2.7. In Vitro Transfection 

The cells were cultured (without antibiotics) in a 96-well plate at a density of 5000 cells/well and incubated for 24 h to obtain 70–80% of confluence prior to the addition of polyplexes/lipoplexes. Before the transfection, the medium was replaced with 100 µL of fresh medium (without serum). The vector/DNA complexes were prepared as follows: 0.5 µg DNA per well mixed in (40 µL) DMEM and an appropriate amount of lipid/NHGs mixed in (40 µL) DMEM without serum. After 5 min incubation at room temperature, the diluted DNA was combined with the diluted lipid/NHGs (total volume: 80 µL) and incubated for 20 min at RT. Complexes were added to each well containing 100 µL of medium without serum and mixed gently by rocking the plate back and forth. (The work with the serum is disclosed in the Appendix A.) After 4 h incubation at 37 °C in a CO_2_ incubator, 10% FBS was added without removing the complexes. After incubation for 48 h, the medium (10% FBS) was replaced. Then, after further incubation for 4 days, transgene expression was tested.

### 2.8. In Vitro Transfection (Passing Method)

The cells were harvested after 6 days of incubation as described above, divided into 6 equal portions, and passed to 6 wells. The wells were analyzed one by one at 48 h time intervals starting from day 8 after incubation with polyplexes. Untreated cells were passed in the same way and used as controls.

### 2.9. In Vivo Analysis 

BALB/c mice, 7–8 weeks old, were purchased from ENVIGO ISRAEL. All animal experiments were reviewed and approved by Bar-Ilan University, and all protocols met the requirements of the local ethical committee of Bar-Ilan University, Israel. All mice were fed and allowed free access to drinking water. The animals were monitored for the total duration of the experiment. Two concentrations of mCherry DNA were used for these experiments (20 and 40 µg). In order to complex 20 and 40 µg of DNA, we used 0.5/1 mg of NHGs. We injected the complexes into mice in two ways: intramuscularly (IM) and subcutaneously (SC). In both cases, we injected into one flank a high concentration and into the other flank a low concentration. Six mice were used for intramuscular injection, five mice for subcutaneous injection, and one mouse for the control, and the experiment was continued for 29 days.

### 2.10. Statistical Analysis

Results are reported as means ± standard deviations (n is the number of independent measurements). The Student’s *t*-test was used to determine significant differences, and probability values of *p* < 0.05 were considered significant.

## 3. Results and Discussion

### 3.1. In Vitro Experiments

#### 3.1.1. Cell Proliferation 

In a separate publication [29], we showed that the NHGs as well as their DNA complexes were devoid of toxicity after 48 h incubation in several cell lines. Herein, we disclose results for HEK293T cell lines and measurements of toxicity at times where transfections were efficient. Unprecedently, we tested toxicity after 6 days of incubation. The results in Figure 1 demonstrate that polyplexes (400, 200, and 50 nm) were devoid of toxicity using NHG/DNA ratios of 20:1 at concentrations of 10 µg:0.5 µg in 100 μL medium in a HEK293T cell line. A standard lipoplex formed with the same amount of DNA (0.5 µg) using cationic lipid RPR-120535 (6 nmol) [30] displayed significant toxicity after 6 days of incubation. The cytotoxic effect values of polyplexes were calculated based on quadruplicates at concentrations of 10 µg NHGs/0.5 µg DNA and 10 µg for naked NHGs. The lipoplexes, naked DNA (0.5 µg GFP), and triton were evaluated as controls.

HeLa and NIH3T3 cells gave similar results. 

#### 3.1.2. In Vitro Transfections 

To investigate the potential of NHGs to carry and deliver DNA, 200 nm NHGs were complexed with pGFP reporter gene plasmids at different ratios and incubated with HEK-293T cells for 6 days. The amount of pDNA was the same for all the complexes (0.5 µg). In Figure 2, we can see from controls A2 (NHGs alone) and A3 (NHGs complexed with salmon sperm DNA) that there were significant numbers of cells and thus that there was no significant toxicity after 6 days of incubation as compared to the non-treated cells in A1. Increasing ratios of NHGs/pDNA from A4 to A7 brought about improved transfections, with the highest rate in A7 at 12.5 µg NHGs. These results are unprecedented, since protein expression after transfections with lipo/polyplexes was mostly observed 2 days after incubation.

To understand the kinetics of the penetration of complexes into the cells, we prepared Cy5-labelled NHGs of 200 nm and followed the cell penetration of their complexes side by side with protein expression over different periods of time. Complexes were formed at different NHG/DNA ratios and analyzed. Here, we show the results for the best ratios for each cell line. Standard cationic lipid RPR120535 was used again as a reference at the known optimal transfection lipid/DNA ratio. Figure 3 shows the results for transfections of HEK293T cells after 24 h, 48 h, and 6 days of incubation. Significant transfections for RP120535 (Figure 3A,D) were observed as compared to NHGs/DNA (Figure 3B,C,E,F). Although the levels of transfection of the NHG complexes were very low at 24 and 48 h, the presence of the complexes inside the cells could be clearly observed in the Cy5 channel (Figure 3C,F). 

We attribute the low transfections observed at 24–48 h to slow release of the DNA from the complexes. Thus, we decided to investigate longer incubation times. Longer incubation times need refreshing supernatants to keep the cells healthy. Thus, in a new panel of experiments, 48 h after incubation the medium was exchanged and incubated for an extra 4 days. Figure 3, panels G–I show the results obtained after 6 days of incubation. Standard RPR120535 lipid/DNA shows a clear decrease in the number of cells, denoting the significant toxicity of the standard transfection agents with fewer cells that were transfected (see Figure 3G). On the other hand, the cells divided normally in panels H and I, both sets displaying strong fluorescence, which indicates high transfection with no cell toxicity. A stock of polyplexes was also present inside the cells in panel I, where we used Cy5-labelled NHGs for transfection. This polyplex stock can induce a continuous release of the transgene over many days. Since the turnover of proteins in cells is about 1–2 days, we suggest that a continuous release and expression of the transgene must be promoted by the presence of the NHG complexes in the cells. The lack of toxicity of the new NHGs allowed the performance of a long-time incubation experiment without interfering with the cell growth. This is the first time that a continuous release of DNA and protein expression has been shown for polyplexes or lipoplexes. The results open the gates for in vivo sustained expression of transgenes using our NHGs. 

HeLa cells were incubated with varying ratios of 200 nm NHGs/DNA. The results indicate a similar behavior as for the HEK293T cells: after 6 days of incubation, lipid complexes were toxic to the cells and good transfections were observed for NHGs with an optimal ratio of 12.5 µg NHGs/0.5 DNA (Figure 4).

To evaluate and compare transfections using different NHG sizes, HEK293T and HeLa cells were treated also with 400 nm polyplexes. The 400 nm NHGs were complexed with GFP at different ratios and incubated for 6 days. The findings demonstrated that HEK293T cells were transfected at lower weight ratios using 400 nm NHGs (Figure 5C). We also noted that transfections with 200 nm NHGs (Figure 5B) were more efficient than transfections using 400 nm NHGs (Figure 5C–F).

As for the HeLa cells, no significant transfection was observed at any weight ratio with 400 nm NHGs.

Interestingly, 50 nm polyplexes disclosed a smaller number of transfected cells in HEK293T cells after 6 days of incubation as compared to 200 nm NHGs (see Appendix A) and no transfections for HeLa cells.

Overall, we have shown that the 200 nm NHGs displayed high transfection efficiency at day 6 after incubation with no significant cell toxicity. The 400 nm NHGs displayed significant transfections but at a lower weight ratio with DNA. Finally, 50 nm NHGs resulted in poor transfections at any weight ratio.

#### 3.1.3. Sustained In Vitro Transfection after Cell Passing Using 200 nm Polyplexes 

We used a passing experiment to analyze the long-term gene transfection in a HEK-293T cell line. The cells were harvested after 6 days of incubation with polyplexes (12.5 µg NHGs 200 nm/0.5 µg DNA) and were divided into six equal portions and passed to six wells. The wells were analyzed one by one at 48 h time intervals starting from day 8 after incubation with polyplexes and appropriate renewal of media over the course of the experiment. Untreated cells were passed in the same way and were used as controls. The results revealed that the polyplexes could efficiently keep the cells transfected 48 h after passing (Figure 6B) and 96 h after passing (Figure 6C). Again, we are able to show evidence that the complexes were inside the cells and that there was a continuous release of DNA; its expression must have been present since protein turnover is only about 24 h.

#### 3.1.4. Semi-Quantitative Analysis of In Vitro Transfections in Various Cell Lines Using an Incucyte Microscope

Various concentrations of NHGs/GFP (400, 200 and 50 nm) were used to analyze the gene transfection in the different cell lines: HEK293T, HeLa, and NIH-3T3. Non-treated cells were used as negative controls, and 6 nmol lipid RPR2120535/GFP was used as a positive control. The expression profiles of cells were analyzed by the normalized green area/phase area. The results indicated that 200 nm polyplexes had better transfection efficiencies in all three cell lines as compared with both 400 and 50 nm polyplexes (Figure 7 and Appendix A). However, with the 400 nm polyplexes, few cells were transfected at lower weight ratios. Moreover, no significant transfection was observed for 50 nm polyplexes in all three cell lines. 

#### 3.1.5. In Vivo Transfections

Our in vitro results demonstrated a long-term expression of the marker gene for 10 days (including cell passing) after incubation without significant cytotoxicity. In vivo experiments using small mammalian rodents are crucial for evaluating the feasibility of local protein expression as a first step towards the development of a new vaccine methodology for the delivery of foreign proteins. The present study aimed to determine the efficiency of the transfection of a fluorescent gene marker (m-cherry plasmid instead of GFP) by local administration of a DNA/NHG formulation that slowly releases the DNA. Two ways of administration were tested: subcutaneous and intramuscular, using two different concentrations of the DNA (20 and 40 µg). The expression of the gene was followed by in vivo fluorescence imaging (using a Maestro camera) for four weeks, with images taken every 2 to 3 days after administration. Finally, after termination, organs were collected and observed directly by the same method, and histology was performed on selected slices from the tissues.

Figure 8 and Figure 9 show m-Cherry expression at the sites of injection for both IM and SC, as observed using the Maestro platform. The expression started to be observed at day 5, became significant at day 12 (*p* < 0.01, n = 6), and remained stable until the end of the experiment at day 29. The first and important conclusion from these observations is that the DNA must remain functional over the course of the experiment, since the turnover of the fluorescent protein is about 24 h. Signals observed at day 29 were the result of transfection occurring 24–48 h before day 29. The NHG/pDNA complex maintains DNA functionality for several weeks; thus, a slow release of the DNA results in long-term expression. This result is concomitant with the observed in vitro sustained gene expression after passing the cells for long periods of time (see Figure 6).

Analysis of the obtained average signals per square centimeter of extracted tissue (Figure 10) revealed that signals generally remained significant for the duration of the experiment and were stronger for SC as compared to IM administration. This is supported by previous work that demonstrated that administered hydrophilic materials undergo better spreading from muscle as compared to fatty tissues in SC administration [31]. We will discuss this issue again while examining the histological slices of organs (see below).

After termination of the experiment, organs were collected and measured under the Maestro camera. One animal from each group was sacrificed in the middle of the experiment (day 14), and their organs were measured (see Figure 11). We clearly observed fluorescence at the sites of administration, but we could also observe significant expression in the brain and heart (Figure 12). Since at this time we did not measure a control animal, we will discuss the in vivo expressions at the end of the termination.

The rest of the animals were sacrificed at day 29, and their organs were collected and analyzed. Clear fluorescence was observed at the site of injection both for IM and SC administration (Figure 13) and, astonishingly, in the brain and heart (see Figure 14) as compared to the non-treated control. A summary of signals in different organs is given in Figure 15, where significant signals can be seen in the brain and heart. The two animals sacrificed at day 14 helped to understand the trend of transfection strength at different times by analyzing the IM and SC average signals for day 14 as compared to day 29 for the different tissues and organs (see Figure 16). We noted that the level of signals was of the same order; thus, we concluded that there was a continuous transfection that kept significant levels of the expressed protein not only in the site of administration but also in the brain and heart.

Overall, the in vivo experiment suggested that there was a continuous release of DNA plasmids, allowing the sustained expression of a foreign protein at the site of administration (either IM or SC) at both concentrations during the 29 days of the experiment. Moreover, somehow, the DNA could be transported to other organs, such as the brain and heart, and the foreign protein was expressed in these organs at lower levels.

In view of the unexpected and striking results obtained for the heart and brain, we were encouraged to perform histological studies of the injected tissues and positive organs. 

The positive tissues were submitted to histological analysis. Thus, the IM and SC administration tissues, as well as the brains and hearts, were fixed and stained with Hematoxylin/Eosin or directly observed with DAPI nuclear staining and direct detection of m-cherry protein. We acknowledge that at the sites of injection, the transfected cells were mostly those of the immune system and not from the local tissues. Figure 17 shows the expression of the m-cherry protein at the sites of administration, both IM (upper panels) and SC (lower panels), as compared to the controls.

The delayed protein expression observed in the in vivo experiments (see Figure 9) can be explained in terms of immune cell recruitment by the complexes. Only after 5 days we observed significant protein expression that remained significant until the end of the experiment. We suggest that this was due to the enrichment of the site with immune cells which were then transfected. 

The brain slices showed expression at the meninges area of m-cherry in immune cells, as observed by H&E staining (see Figure 18).

The heart slices showed expression of mcherry in immune cells, as observed by H&E staining (see Figure 19). 

The histological studies showed the presence of the mcherry protein not only at the sites of administration (IM and SC) but also in remote organs, such as the heart and brain. We acknowledge the infiltration of immune cells in the injected tissues, and the histology confirmed that the cells expressed the mCherry protein. We propose that the NHG/DNA complexes were internalized by these immune cells. Immune cells are known to be capable of migrating to different parts of the body; thus, we suggest that our NHG/DNA complexes act as Trojan Horses: they are internalized by immune cells recruited to the site of administration. These loaded cells migrate throughout the body and reach the heart and the brain, and there they express the foreign protein. The site of administration acts as a pool for loading the Trojan Horses into immune cells which then migrate to other organs. All this can easily happen and be observed due to the lack of toxicity of the complexes. 

We speculate and suggest that the presence of the foreign protein in the heart is the first experimental evidence for the myocarditis/pericarditis observed in some children after COVID-19 vaccination using mRNA. We suggest that some of the spike protein is expressed in the heart area by immune cells that migrate from the site of administration and that this mediates the inflammation of the organ.

## 4. Conclusions

In this study, we successfully applied various sizes of reduced NHGs and their polyplex complexes with pDNA to gene delivery and protein expression with high biocompatibility. This unprecedented biocompatibility allowed us to observe effects that could not be followed before using lipoplexes or polyplexes due to their cell toxicity. The in vitro results revealed that the polyplexes showed a delayed protein expression of 6 days but that once the expression started it was maintained significantly after cell passing. The nanohydrogels developed here are the first non-viral systems for gene delivery that are able to deliver DNA in vitro over a long period of time. The in vivo IM and SC administration of m-Cherry/NHG polyplexes indicated a delayed but prolonged expression of the marker gene at the area of administration. A slow release of the DNA and its expression was clearly demonstrated in vitro by the cell passing experiments and in vivo by the prolonged expression for 29 days. The new platform might be applied to vaccination, since it will certainly better protect m-RNA and allow longer expression of the antigenic protein for more efficient m-RNA vaccination. Astonishingly, we noted low but significant protein expression in remote organs, such as the brain and heart, as compared to non-treated organs. After histological analysis, we observed that immune cells were recruited to the sites of injection and underwent transfection. Thus, the NHG/DNA complexes must have penetrated these cells. Since immune cells can migrate, expression in remote organs is possible. The results from histological analysis of brain and heart tissues allow us to speculate that immune cells containing the NHG/DNA complexes migrated to these organs and were able to express the m-cherry protein; thus, we suggest that our complexes behave as Trojan Horses. We speculate that the Trojan Horse effect shown here is the first in vivo evidence for the mechanism of myocarditis observed in some children after m-RNA vaccination against coronavirus [32,33]. We are further studying this migration/expression phenomenon, and results will be presented elsewhere.

## Figures and Tables

**Figure 1 pharmaceutics-15-00961-f001:**
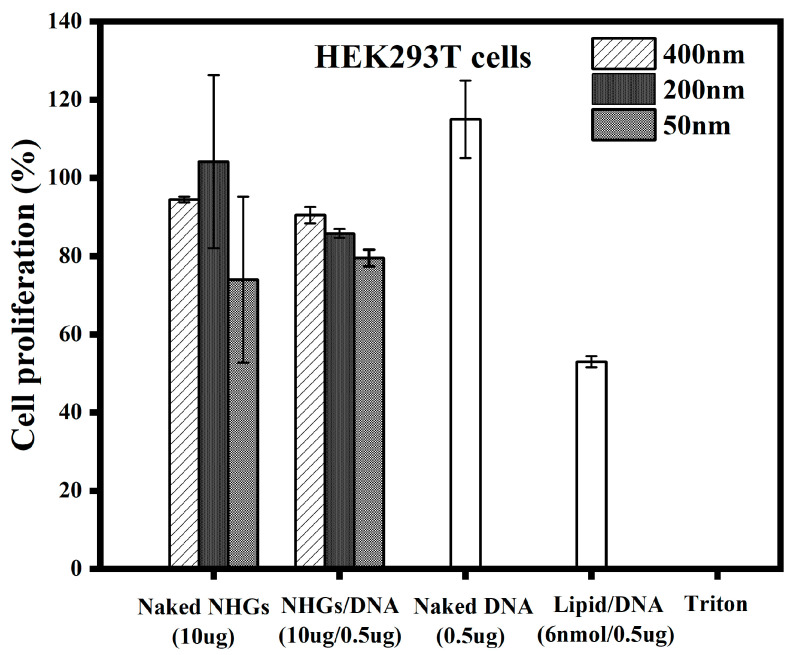
Cell proliferation for the different concentrations of polyplexes (400, 200, and 50 nm) and lipoplexes at 6 days of incubation in HEK293T cells. Each point represents a mean value ± SD (*n* = 4).

**Figure 2 pharmaceutics-15-00961-f002:**
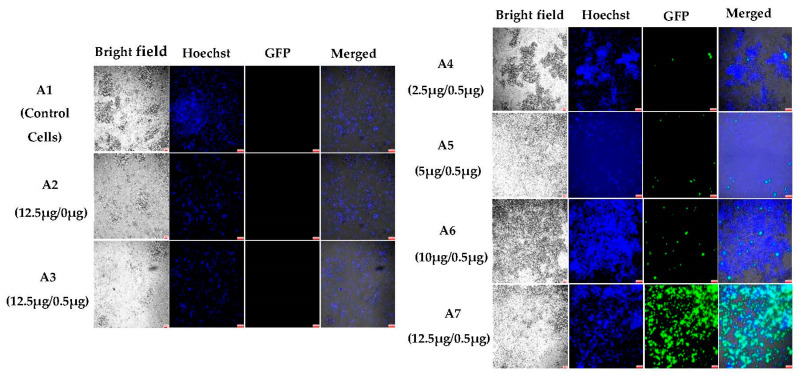
Fluorescent microscopic images (10×) of polyplexes of 200 nm NHGs in HEK293T cells after 6 days of incubation: (**A1**) control cells; (**A2**) 12.5 μg NHGs alone; (**A3**) 12.5 μg NHGs/0.5 μg salmon DNA; (**A4**) 2.5 μg NHG/0.5 μg DNA; (**A5**) 5 μg NHG and 0.5 μg DNA; (**A6**) 10 μg NHG and 0.5 μg DNA; and (**A7**) 12.5 μg NHG and 0.5 μg DNA. (Scale bars for all pictures: 100 μm).

**Figure 3 pharmaceutics-15-00961-f003:**
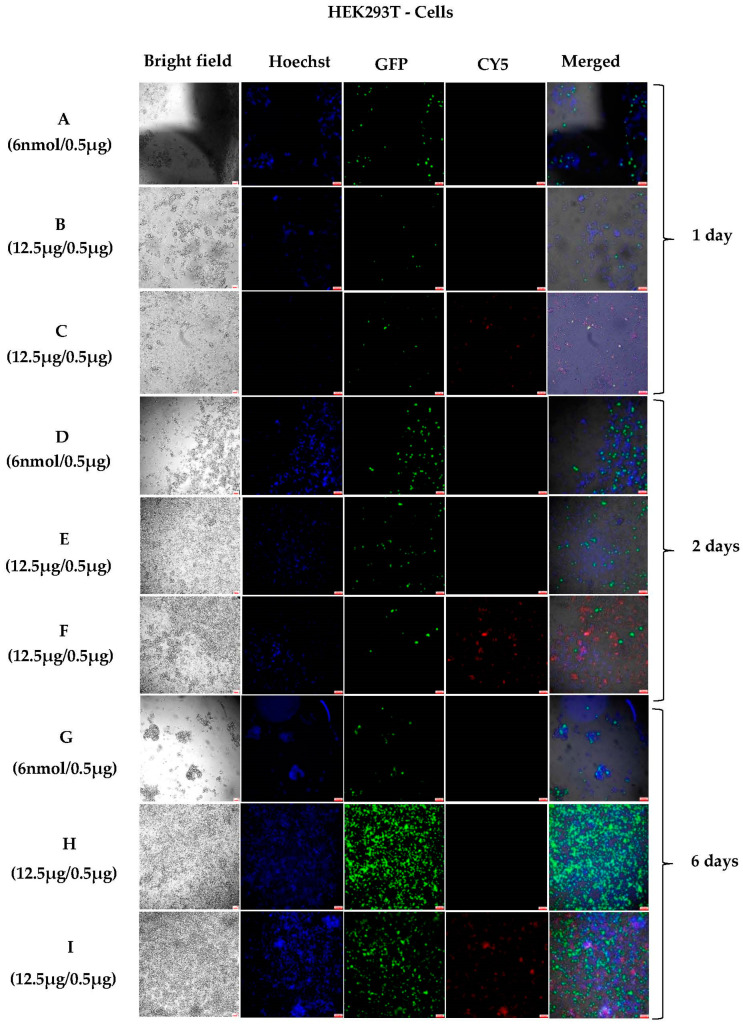
Fluorescence microscopic images (10×) of polyplexes with HEK-293T cells: (**A**,**D**,**G**) 6 nmol lipid/0.5 μg DNA; (**B**,**E**,**H**) 12.5 μg NHG/0.5 μg DNA; and (**C**,**F**,**I**) 12.5 μg Cy5@NHG and 0.5 μg DNA. (Scale bars for all pictures: 100 μm).

**Figure 4 pharmaceutics-15-00961-f004:**
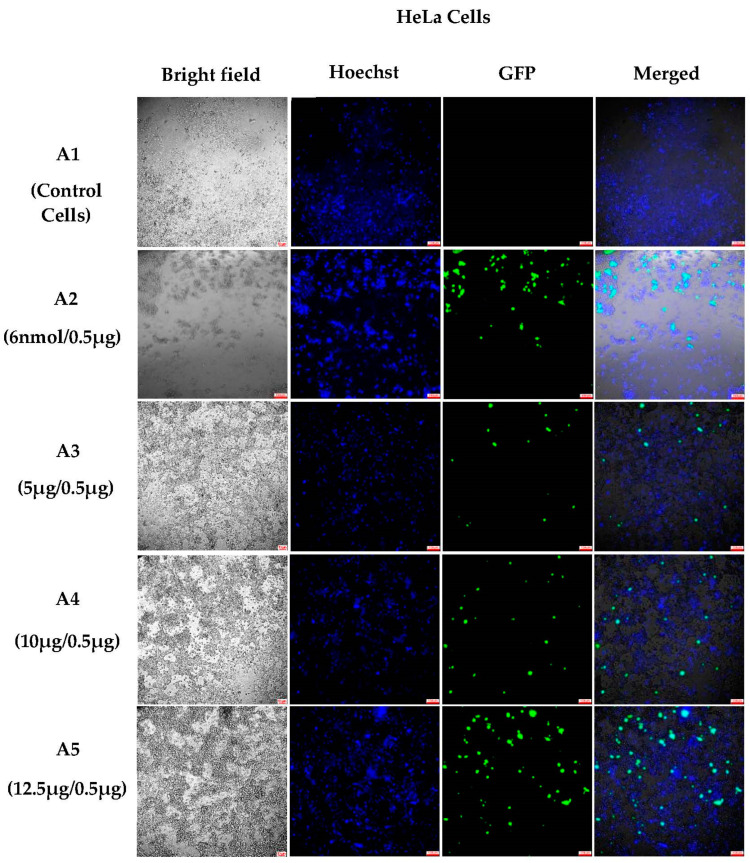
Fluorescent microscopic images (10×) of polyplexes 200 nm with HeLa cells (after 6 days of incubation): (**A1**) control cells; (**A2**) 6 nmol lipid/0.5 μg DNA; (**A3**) 5 μg NHGs/0.5 μg DNA; (**A4**) 10 μg NHGs/0.5 μg DNA; and (**A5**) 12.5 μg NHGs/0.5 μg DNA. (Scale bars for all pictures: 100 μm).

**Figure 5 pharmaceutics-15-00961-f005:**
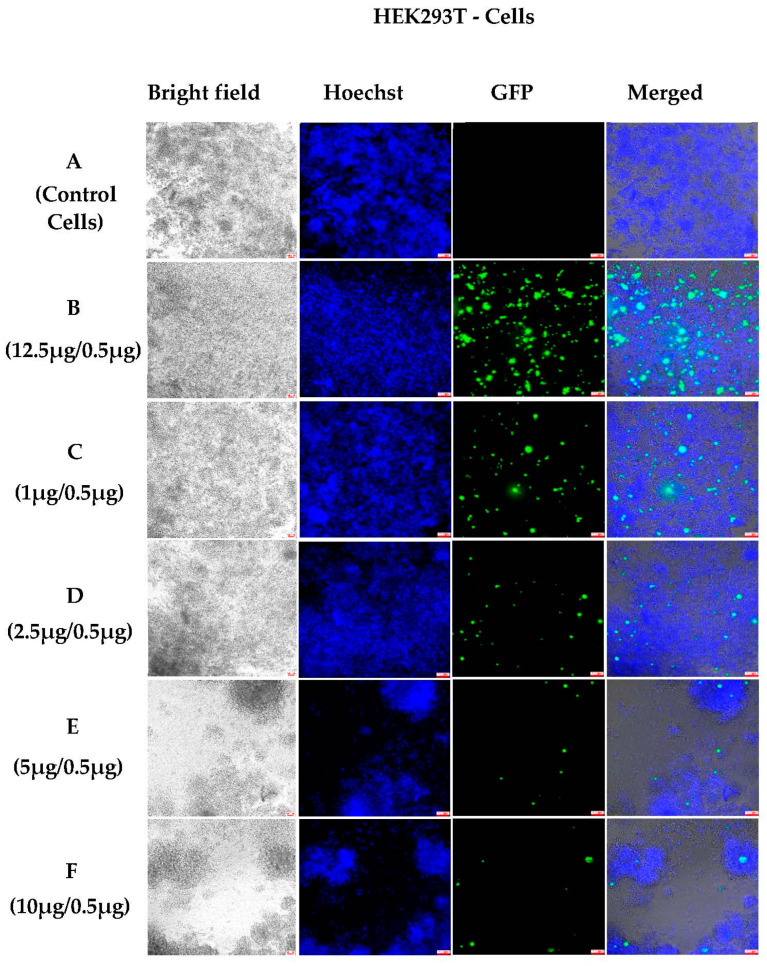
Fluorescence microscopic images (10×) of 400 nm polyplexes in HEK293T cells after 6 days of incubation: (**A**) control cells; (**B**) NHG/pGFP: 12.5 μg/0.5 μg (200 nm control); (**C**) NHG/pGFP: 1 μg/0.5 μg (400 nm); (**D**) NHG/pGFP: 2.5 μg/0.5 μg (400 nm); (**E**) NHG/pGFP: 5 μg/0.5 μg (400 nm); and (**F**) NHG/pGFP: 10 μg/0.5 μg (400 nm). (Scale bars for all pictures: 100 μm).

**Figure 6 pharmaceutics-15-00961-f006:**
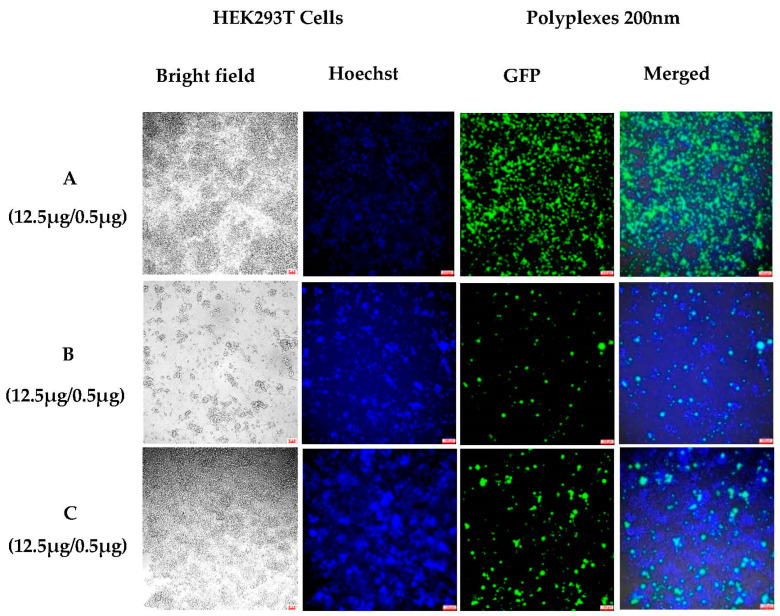
Fluorescent microscopic images (10×) of polyplexes: (**A**) 12.5 μg NHGs with 0.5 μg GFP, after 6 days of incubation; (**B**) 12.5 μg NHGs with 0.5 μg GFP, 48 h after passing; and (**C**) 12.5 μg NHGs with 0.5 μg GFP, 96 h after passing. (Scale bars for all pictures: 100 μm).

**Figure 7 pharmaceutics-15-00961-f007:**
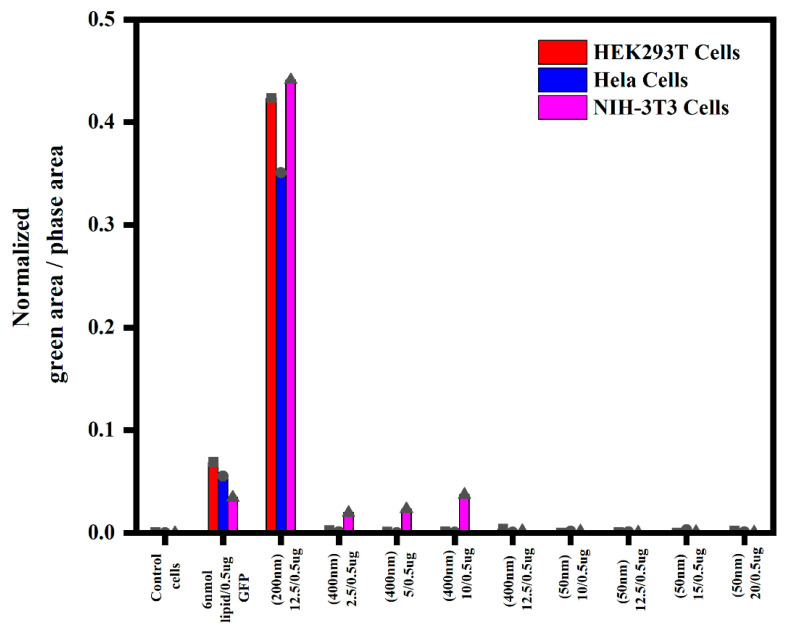
Quantification of fluorescence obtained with an Incucyte microscope for polyplexes (400, 200, and 50 nm) with three cell lines after 6 days of incubation. (See Incucyte pictures in Appendix A; total area for each picture: 2.27 mm^2^.) Each point represents the maximum value of normalized green area/phase area.

**Figure 8 pharmaceutics-15-00961-f008:**
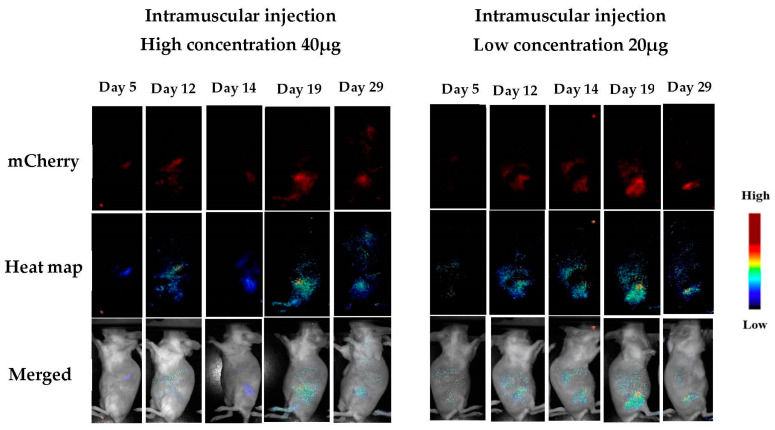
Detection of m-Cherry after intramuscular administration. (Results for days 8, 22, and 26 can be found in the Appendix A).

**Figure 9 pharmaceutics-15-00961-f009:**
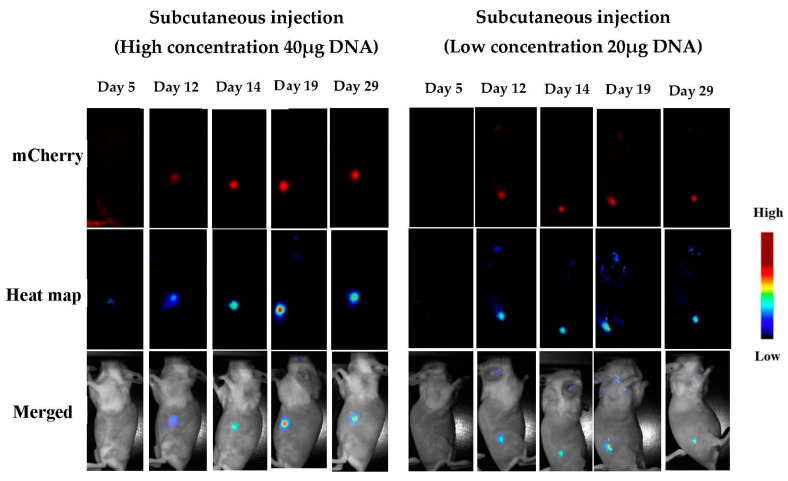
Detection of m-Cherry after subcutaneous administration. (Results for days 8, 22, and 26 can be found in the Appendix A).

**Figure 10 pharmaceutics-15-00961-f010:**
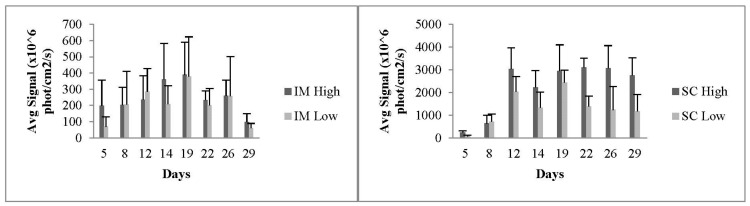
Average signals in vivo as obtained from IM (**left**) and SC (**right**) injected animals. The results are presented as means ± SDs (n = 6), and all statistical analyses (comparisons between the groups and the days) were performed using paired *t*-tests.

**Figure 11 pharmaceutics-15-00961-f011:**
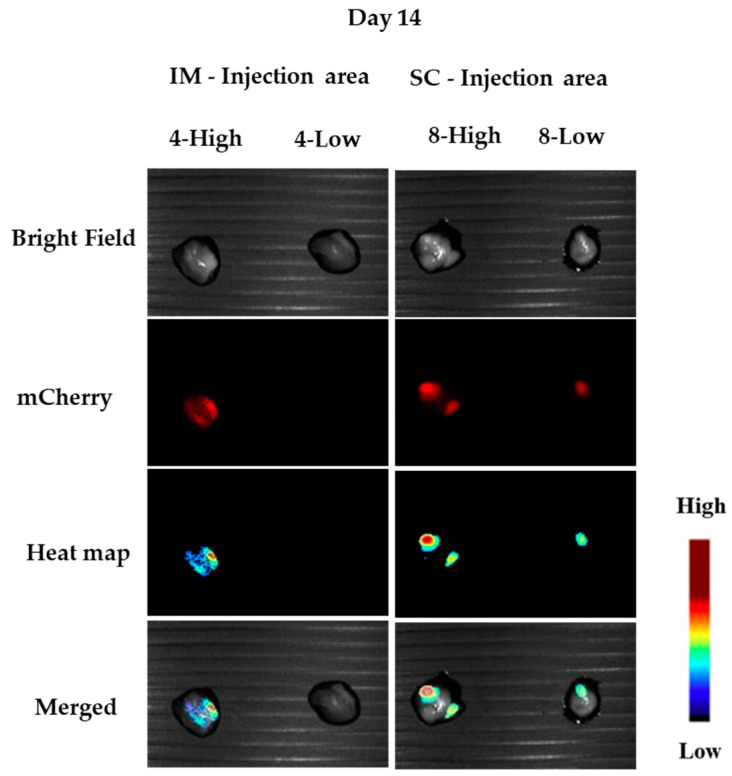
Injection tissues at day 14 (mice 4 and 8): left: IM tissue; right: SC tissue. (High: 40 µg mCherry; Low: 20 µg mCherry).

**Figure 12 pharmaceutics-15-00961-f012:**
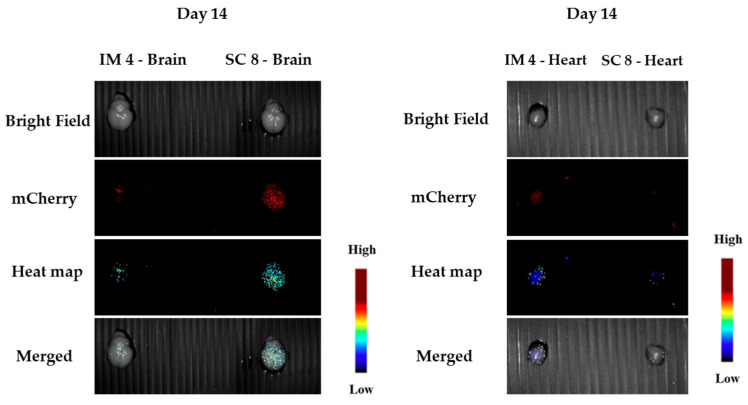
Brain and heart: IM (**left**) and SC (**right**) at day 14.

**Figure 13 pharmaceutics-15-00961-f013:**
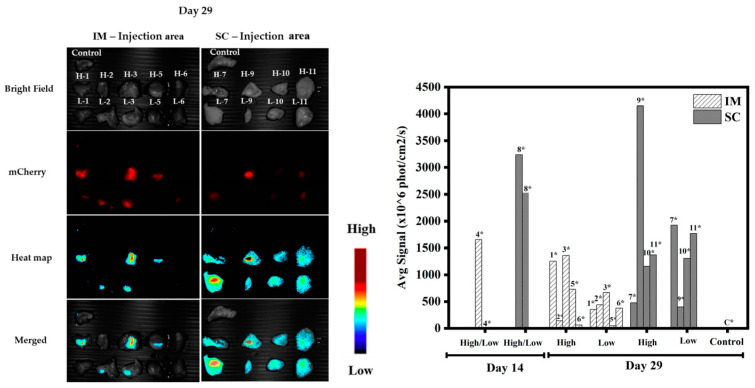
Injected tissues at day 29: (**left**) IM tissue; (**right**) SC tissue. (H1 to H11: 40 µg mCherry; L1 to L11: 20 µg mCherry.) The graph shows the average signals of tissues at days 14 and 29; each point represents the maximum value of the average signal. * Animal number.

**Figure 14 pharmaceutics-15-00961-f014:**
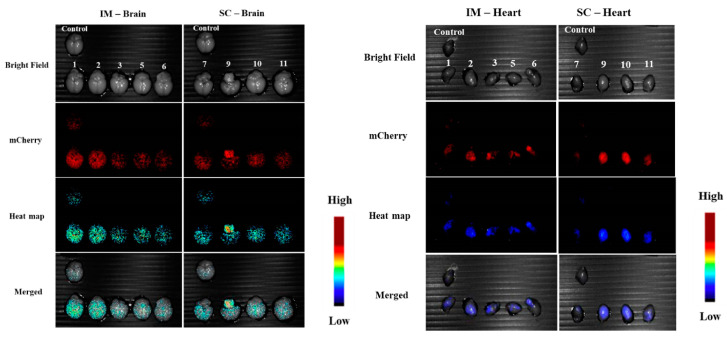
Brain and heart: IM (**left**) and SC (**right**) at day 29.

**Figure 15 pharmaceutics-15-00961-f015:**
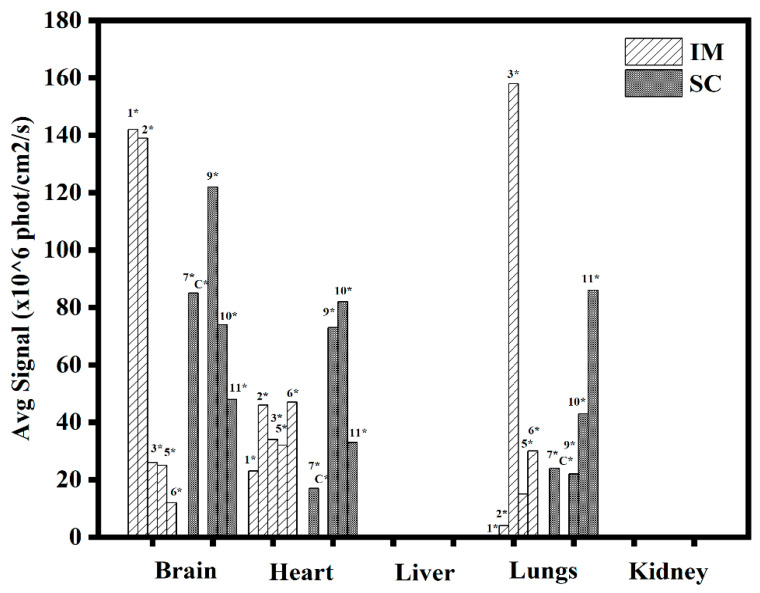
Summary of signals for IM and SC collected tissues at days 29. * Animal number. Controls were negative (C*). Each point represents the maximum value of the average signal.

**Figure 16 pharmaceutics-15-00961-f016:**
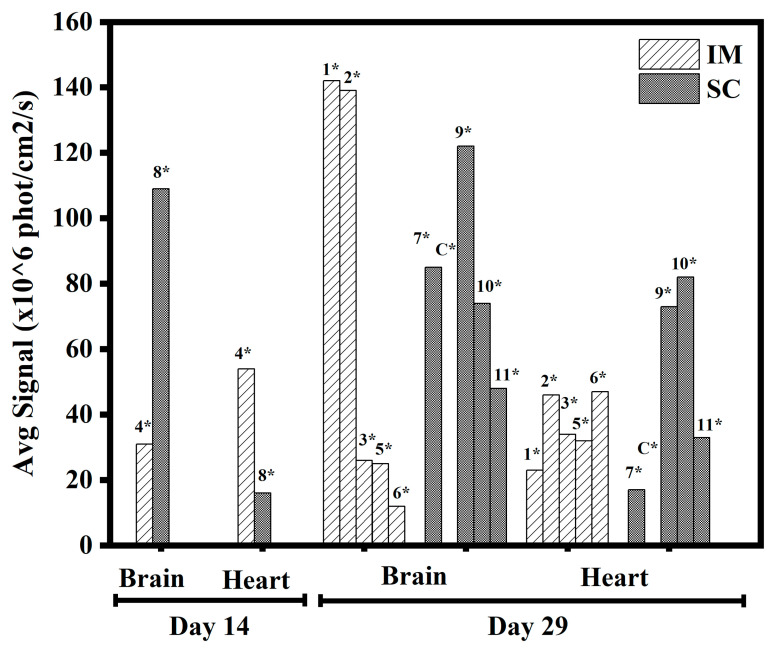
Comparison of IM and SC brain and heart at days 14 and 29. * Animal number (C* control). Each point represents the maximum value of the average signal.

**Figure 17 pharmaceutics-15-00961-f017:**
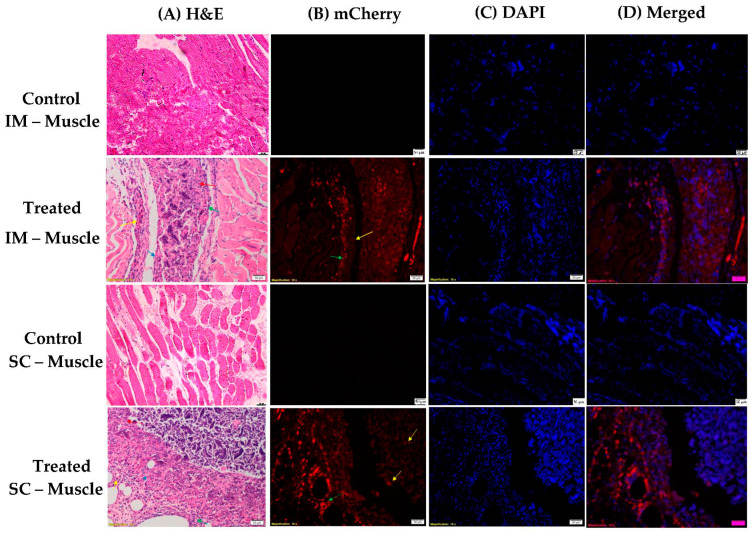
Histological analysis of injected IM muscle (two upper panels) and injected SC tissue (two lower panels). (**A**) Focal extensive infiltrated lesion composed of mineralized material, surrounded by macrophages (red arrows), multinucleated giant cells (blue arrows), lymphocytes (green arrows), and neutrophils (yellow arrows). (**B**) mCherry channel showing focal extensive infiltrated lesion composed of mineralized material (yellow arrows) that is partly fluorescent-positive surrounded by inflammatory cells (green arrow) that are positive. (**C**) DAPI showing all nuclear nuclei in the sample. (**D**) mCherry and DAPI merged. (Scale bars for all pictures: 50 µm).

**Figure 18 pharmaceutics-15-00961-f018:**
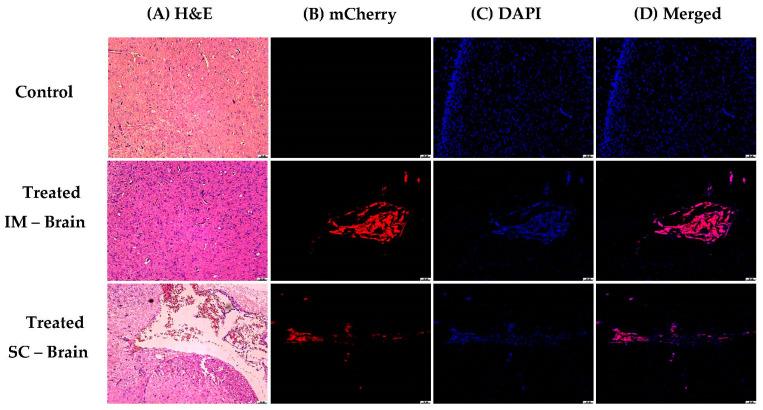
Histological analysis of brain tissues after IM administration. (**A**) Eosin color shows inflammatory cells in the meninges area after both IM (middle panel) and SC (bottom panel) administration. (**B**) Inflammatory cells expressing m-cherry and DAPI (**C**) for IM and SC administration. (**D**) Merging of mCherry/DAPI in pink. Red fluorescent cells not seen with blue DAPI are erythrocytes. (Scale bars for all pictures: 50 µm).

**Figure 19 pharmaceutics-15-00961-f019:**
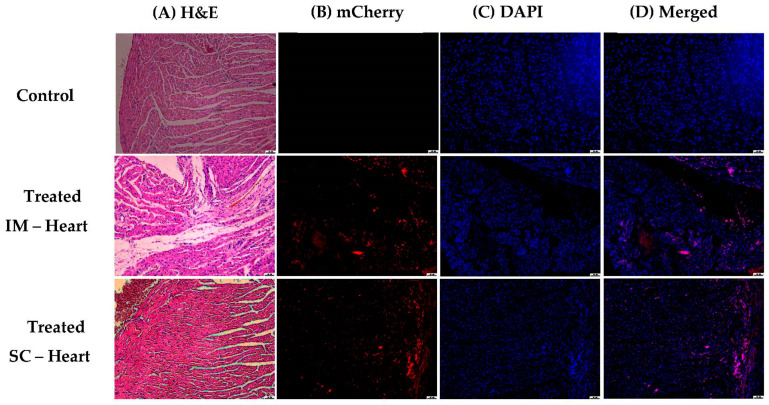
Histological analysis of heart tissues after IM administration. (**A**) H & E coloring showing inflammatory cells in rows 2 and 3. (**B**) Shows cells expressing m-cherry and (**C**) DAPI. (**D**) Merging of m-cherry/DAPI in pink. Red fluorescent cells not seen with blue DAPI are erythrocytes. (Scale bars for all pictures: 50 µm).

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
