# Peer review of "A New Strategy for Nucleic Acid Delivery and Protein Expression Using Biocompatible Nanohydrogels of Predefined Sizes"

_pharmaceutics, 2023, doi:10.3390/pharmaceutics15030961_

Round 1

Reviewer 1 Report

It is an interesting work showing the application of the material that Authours previously synthesized and characterized. In my opinion, it can be considered for publication after the correction of the interpunction mistakes e.g. lack of space between the number and unit (e.g. 4ml instead of 4 mL), ml instead of mL, 5 minutes should be 5 min, please remove double spaces, naked would be better to use to "bare" word, there is Hela instead of HeLa, etc.

Author Response

Reviewer 1:

It is an interesting work showing the application of the material that Authours previously synthesized and characterized. In my opinion, it can be considered for publication after the correction of the interpunction mistakes e.g. lack of space between the number and unit (e.g. 4ml instead of 4 mL), ml instead of mL, 5 minutes should be 5 min, please remove double spaces, naked would be better to use to "bare" word, there is Hela instead of HeLa, etc.

Thank you for the suggestion, we have revised the values from 4ml to 4 mL and 1ml to 1 mL (see lines 80, 81). 5 minutes to 5 min and 20 minutes to 20 min (see lines 128 and 130). Modified from Hela to HeLa (see lines 260, 358, 362, 417, and 473).

Reviewer 2 Report

In present manuscript the authors developed of new nanohydrogels complexed with DNA for effective and safe DNA delivery in vitro and in vivo. The topic of a study - the development of effective DNA/RNA carriers - is new and promising. At present time many researchers are trying to develop such carriers using nanoparticles, hydrogels, polymers, etc. The main achievement of the authors that the nanohydroges presented showed themselves as effective and non-toxic carriers with prolonged in time protein expression. The results of study are correctly done and well described. The manuscript is excellently illustrated by figures. The conclusions are supported by the data. As for English, I am not native speaker, for me English is acceptable. Everything is clear to read and understand. I did not find any serious mistakes in experiments to mention.

I have minor remarks:

1) The description of statistical analysis performed should be included in "Materials and Methods".

2) The statistical description should be included in Figs. 7,10, 13, 16.

3) The Fig. 1 and the corresponding data should be moved to Trash. n=2 is nothing from the point of statistical analysis. S.D. at n=2 looks very funny.

Author Response

Reviewer 2:

In present manuscript the authors developed of new nanohydrogels complexed with DNA for effective and safe DNA delivery in vitro and in vivo. The topic of a study - the development of effective DNA/RNA carriers - is new and promising. At present time many researchers are trying to develop such carriers using nanoparticles, hydrogels, polymers, etc. The main achievement of the authors that the nanohydroges presented showed themselves as effective and non-toxic carriers with prolonged in time protein expression. The results of study are correctly done and well described. The manuscript is excellently illustrated by figures. The conclusions are supported by the data. As for English, I am not native speaker, for me English is acceptable. Everything is clear to read and understand. I did not find any serious mistakes in experiments to mention.

I have minor remarks:

  • The description of statistical analysis performed should be included in "Materials and Methods".

Thank you for the suggestion, we have added the description of statistical analysis in materials and methods (see lines from 156 to 158).

  • The statistical description should be included in Figs. 7, 10, 13, 16.

 We have included the details, thank you.

  • The Fig. 1 and the corresponding data should be moved to Trash. n=2 is nothing from the point of statistical analysis. S.D. at n=2 looks very funny.

Sorry for this mistake, we have added the details of statistical analysis in material and methods, and n=4 represents number of independent experiment. Additionally, more data on toxicity appears in the back to back publication reference 29

Reviewer 3 Report

The manuscript by Eswaran et al describes the “follow up” characterization of a new nanohydrogel system for the delivery of plasmid DNA which was recently published.

Generally, this is a very interesting system and with great potential for further studies. The manuscript is written in a logical order and easy to follow.

To begin, it would be very helpful if the authors can again very briefly describe the NHG system or adding some structures in the supplement. The reader has absolutely no idea what compounds were used or kind of structure the NHG have and needs to look it up in the previous publication.

The used NHG system can be prepared in different size ranges of 50, 200 and 400 nm and were tested at different ratios and different cell lines. In fact, the nanoparticles seem to be very biocompatible as 0.5µg pDNA per 96 well is a huge amount also used with the high mass ratio for the NHG. A very interesting finding, which I would not have expected for these NPs, is the sustained release behavior. The reporter gene expression in vitro was steadily increasing for the first six days and remained constant for at least 10 days!

For the in vitro transfections, there is one point I do not like. The transfections were for the first 4 hours in the absence of serum. Do the authors know if the complexes are not stable in serum? This is a point that must be addressed.

Another interesting point is the size-dependence of the NHGs and the transfection efficiencies. The 200 nm particles worked best while for the 400 nm a general decrease in transfection efficiency was observed but the optimal mass ratio changed towards lower NHG/pDNA ratios. In contrast, the 50 nm complexes have shown no significant transfection.

Based on the in vitro results, the 200 nm NHG were used for in vivo transfections. The complexes were injected either intramuscularly or subcutaneously with two different pDNA dosages. With both injection routes mCherry signals were observed about 5 days later until the end of the study (after 29 days) This is a very impressive result! Thereafter, the authors checked the different organs for mCherry expression and found signals in the brain and heart. These organs are generally very difficult to transfect esp. after i.m. or s.c. injections. The authors performed histological analysis of the organs and found that the mCherry expression was in immune cells and not in the specific tissue. This is absolutely conclusive!

It would be quite interesting for the future if the 50 nm NHGs perform even better due to a deeper tissue penetration and if the transfection of organ cells can be achieved by other injection routes like intravenously or intraperitoneally!

Beside the very interesting results, some points should be improved or addressed as well:

The brightfield pictures of the microscopic images are for all figures too dark. It is absolutely difficult to see the cells.

In the figures 3, 4 the assignments do not fit with the text. For example: line 233 GFP expression in A4 and B4 is the Cy5 panel or I am wrong? In figure 4 no description fits to the text. Also, the “polyplexes 200nm” above the pictures is somewhat misleading and can be moved to the left side next to “A3”.

In figure 13: Are the mCherry and heat map images displayed with the same settings for both injection routes? It seems that sample “L-7 is brighter in the heat map as compared to the mCherry images and heat maps for samples H-6 or L-3?

Author Response

Reviewer 3

The manuscript by Eswaran et al describes the “follow up” characterization of a new nanohydrogel system for the delivery of plasmid DNA which was recently published. Generally, this is a very interesting system and with great potential for further studies. The manuscript is written in a logical order and easy to follow. To begin, it would be very helpful if the authors can again very briefly describe the NHG system or adding some structures in the supplement. The reader has absolutely no idea what compounds were used or kind of structure the NHG have and needs to look it up in the previous publication.

Answer: The NHGs used here appear in a back to back publication reference 29 in the very same issue of this journal, it is not difficult to open that article and check the properties of the nanohydrogels. For the sake of having manuscript with appropriate number of pages the work was splitted into two back to back articles sent to the very same special issue, thus adding here data on the nanohydrogels is rather superfluous.

The used NHG system can be prepared in different size ranges of 50, 200 and 400 nm and were tested at different ratios and different cell lines. In fact, the nanoparticles seem to be very biocompatible as 0.5µg pDNA per 96 well is a huge amount also used with the high mass ratio for the NHG. A very interesting finding, which I would not have expected for these NPs, is the sustained release behavior. The reporter gene expression in vitro was steadily increasing for the first six days and remained constant for at least 10 days! For the in vitro transfections, there is one point I do not like. The transfections were for the first 4 hours in the absence of serum. Do the authors know if the complexes are not stable in serum? This is a point that must be addressed.

Thanks for the comment, we did check the nanohydrogels in the presence of serum from the beginning and they work perfectly (the results were added as supplementary file see figure S3 and now is noted in line 133 of the manuscript). Why we preferred to put in the front the results without serum? The reason is the comparative studies with the control lipid that can only be used in the absence of serum during 4 h. We wanted to compare as close as possible the conditions for both systems.

Another interesting point is the size-dependence of the NHGs and the transfection efficiencies. The 200 nm particles worked best while for the 400 nm a general decrease in transfection efficiency was observed but the optimal mass ratio changed towards lower NHG/pDNA ratios. In contrast, the 50 nm complexes have shown no significant transfection. Based on the in vitro results, the 200 nm NHG were used for in vivo transfections. The complexes were injected either intramuscularly or subcutaneously with two different pDNA dosages. With both injection routes mCherry signals were observed about 5 days later until the end of the study (after 29 days) This is a very impressive result! Thereafter, the authors checked the different organs for mCherry expression and found signals in the brain and heart. These organs are generally very difficult to transfect esp. after i.m. or s.c. injections. The authors performed histological analysis of the organs and found that the mCherry expression was in immune cells and not in the specific tissue. This is absolutely conclusive! It would be quite interesting for the future if the 50 nm NHGs perform even better due to a deeper tissue penetration and if the transfection of organ cells can be achieved by other injection  routes like intravenously or intraperitoneally! Beside the very interesting results, some points should be improved or addressed as well: The brightfield pictures of the microscopic images are for all figures too dark. It is absolutely difficult to see the cells.

Thank you for the suggestion:  we have tuned all bright field pictures across the manuscript and improved some pictures (histology slides that had some artifactual yellow in the back).

 In the figures 3, 4 the assignments do not fit with the text. For example: line 233 GFP expression in A4 and B4 is the Cy5 panel or I am wrong?

We have revised the text (see lines 237-239, 241, 246, 249 and 252), thank you.

In figure 4 no description fits to the text. Also, the “polyplexes 200nm” above the pictures is somewhat misleading and can be moved to the left side next to “A3”.

We have revised the text and removed the ‘Polyplexes 200 nm’ from the above picture as noted in the line 358.

In figure 13: Are the mCherry and heat map images displayed with the same settings for both injection routes? It seems that sample “L-7 is brighter in the heat map as compared to the mCherry images and heat maps for samples H-6 or L-3?

We have used the same setting for both injection routes. When we observe directly the mCherry signals we can see only the strong areas, but with the heatmap twe see also the scattered light around as blue halo.

Reviewer 4 Report

Lakshmanan et al. described the work on biocompatible nanohydrogels/DNA complexes for gene delivery and foreign protein expression. The manuscript showed a bunch of data to show the transfection effect. However, many of the figures showed low quality, which lowered my interest on the work and raised my concerns on whether the approach was available.

 1. The authors claimed that they designed and synthesized new biocompatible nanohydrogels/DNA complexes. However, I did not find any physical and chemical characterization of the nanohydrogels or nanohydrogels/DNA complexes. Furthermore, what is the novelty in the nanohydrogels?

 2. Figure 1, why is the NHGs dosage of group “naked NHGs” 20 mg instead of 10 mg the same as that of group “NHGs/DNA”? The unit used in the manuscript was wrong.

 3. Figure 2, the brightness of the group “bright field” is not enough. The magnification of each picture is too small and the image is fuzzy. Each group should be marked with the concentration or dosage to make it more clear instead of A1, A2, etc. The same problems were also showed in Figure 3, 4, and 5.

 4. Figure 8, the observation time seems to be irregular. Why chose these days to observe? The same problems were also showed in Figure 9 and 10.

 5. Figure 12, the fonts and sizes of the two pictures are inconsistent.

 6. Figure 17, the scale bars of group (B), (C) and (D) are too small, and the images are fuzzy. Group (D) should be marked "merged".

Author Response

Reviewer 4:

Lakshmanan et al. described the work on biocompatible nanohydrogels/DNA complexes for gene delivery and foreign protein expression. The manuscript showed a bunch of data to show the transfection effect. However, many of the figures showed low quality, which lowered my interest on the work and raised my concerns on whether the approach was available.

Thanks for the comment, we have improved all the figures.

  1. The authors claimed that they designed and synthesized new biocompatible nanohydrogels/DNA complexes. However, I did not find any physical and chemical characterization of the nanohydrogels or nanohydrogels/DNA complexes. Furthermore, what is the novelty in the nanohydrogels?

We have showed the physical and chemical characterization of the nanohydrogels or nanohydrogels/DNA complexes in the back to back article appearing in the very same special issue. WE ehanged the last paragraph of the abstract to prevent misunderstandings see lines 19-21 in the manuscript.

Reference: Eswaran, L.; Kazimirsky, G.; Byk, G. New Biocompatible Nanohydrogels of Predefined Sizes for Complexing Nucleic Acids. Pharmaceutics 2023, 15(2), 332; https://doi.org/10.3390/pharmaceutics15020332.

The novelty of the system is that, we have developed a new generation of NHGs by the introduction of new monomers and methodologies that generate NHGs suitable for complex nucleic acids as potential tools for gene delivery and foreign protein expression.

  1. Figure 1, why is the NHGs dosage of group “naked NHGs” 20 mg instead of 10 mg the same as that of group “NHGs/DNA”? The unit used in the manuscript was wrong.

We have revised the figure 1, thank you.

  1. Figure 2, the brightness of the group “bright field” is not enough. The magnification of each picture is too small and the image is fuzzy. Each group should be marked with the concentration or dosage to make it more clear instead of A1, A2, etc. The same problems were also showed in Figure 3, 4, and 5.

We have tuned the pictures of bright field across the manuscript and also added the concentration details in figures. Thank you.

  1. Figure 8, the observation time seems to be irregular. Why chose these days to observe? The same problems were also showed in Figure 9 and 10.

We have omitted other days due to the space limitation. However, we have added the results of other days in supplementary information (see figure S4 and S5).  Days 8, 22 and 26.

  1. Figure 12, the fonts and sizes of the two pictures are inconsistent.

We have revised the figure 12, thank you.

  1. Figure 17, the scale bars of group (B), (C) and (D) are too small, and the images are fuzzy. Group (D) should be marked "merged"

We have revised the data and noted the scale bar in the captions (see yellow highlighted in the manuscript), thank you.

Round 2

Reviewer 1 Report

Thank you for addressing all comments and correcting the manuscript.

Author Response

Thank you, no modifications were further requested by this reviewer